# KRT6A and KRT17 Mark Distinct Stem Cell Populations in the Adult Palpebral Conjunctiva and Meibomian Gland

**DOI:** 10.3390/cells14241979

**Published:** 2025-12-12

**Authors:** Xuming Zhu, Mingang Xu, David M. Owens, Sarah E. Millar

**Affiliations:** 1Black Family Stem Cell Institute, Icahn School of Medicine at Mount Sinai, New York, NY 10029, USA; 2Institute for Regenerative Medicine, Icahn School of Medicine at Mount Sinai, New York, NY 10029, USA; 3Department of Cell, Developmental and Regenerative Biology, Icahn School of Medicine at Mount Sinai, New York, NY 10029, USA; 4Department of Dermatology, Columbia University, New York, NY 10032, USA; 5Department of Pathology & Cell Biology, Columbia University, New York, NY 10032, USA; 6Department of Dermatology, Icahn School of Medicine at Mount Sinai, New York, NY 10029, USA; 7Department of Oncological Sciences, Icahn School of Medicine at Mount Sinai, New York, NY 10029, USA

**Keywords:** KRT6A, KRT17, meibomian gland, conjunctiva, HDAC3, KLF4, GLI2

## Abstract

Purpose: This study aims to investigate whether two stress keratins, KRT6A or KRT17, label self-renewing stem cells (SCs) in adult mouse Meibomian gland (MG), the palpebral conjunctiva (PC) homeostasis, and to explore the mechanisms regulating their expression. Methods: KRT6A and KRT17 expression in adult mouse MG and PC were examined by single-nucleus RNA sequencing and immunofluorescence (IF). Lineage-tracing experiments were performed using *Krt6a-Cre^ERT2^* and *Krt17-Cre^ERT2^* mice carrying the *Rosa26R^nTnG^* or *Rosa26R^mTmG^* reporter. As Hedgehog (Hh) signaling, the histone deacetylase HDAC3, and the transcription factor KLF4 regulate KRT6A and KRT17 in other contexts, IF was conducted to assess the in vivo effects of overexpression of the Hh pathway activator GLI2ΔN, and inducible epithelial deletion of *Hdac3* or *Klf4* on KRT6A and KRT17 expression in the MG and PC. Results: KRT6A and KRT17 are primarily expressed in the MG central duct and ductules. KRT6A also shows robust expression in PC. Lineage tracing indicated that *Krt17* labels self-renewing SCs in the MG, whereas *Krt6a* labels SCs in the PC. GLI2ΔN overexpression induced ectopic KRT17 expression in MG acini and PC but did not affect KRT6A expression in either MG or PC. *Hdac3* deficiency caused expanded expression of KRT6A and KRT17 in MG acini, ectopic KRT17 expression in PC, and increased KRT6A expression in PC basal layer. *Klf4* deletion resulted in ectopic KRT17 expression in PC but did not influence KRT6A expression in MG or PC. Conclusions: *Krt6a*- and *Krt17*-expressing cells contribute to adult PC and MG homeostasis, respectively. KRT17 expression is enhanced by GLI2ΔN, and suppressed by HDAC3 and KLF4, whereas KRT6A expression is controlled only by HDAC3. These findings provide important biological insight into tissue-specific maintenance mechanisms and may inform future therapeutic strategies for regenerating MG and PC tissues affected by SC exhaustion or dysregulation.

## 1. Introduction

The tear film is a dynamic, multi-layered structure with a complex composition that includes water, electrolytes, mucins, proteins, and lipids [1]. It is composed of three distinct layers: an outer lipid layer, a central aqueous layer, and an inner mucin layer that interfaces with the conjunctiva [2]. Beyond its role in hydrating and lubricating the ocular surface, the tear film is essential for maintaining ocular comfort, protecting against infection, reducing inflammation, facilitating wound healing, removing debris, and preserving clear vision [1].

Tear film lipids are provided by meibum, which is secreted from specialized sebaceous glands that are embedded within the tarsal plate and are known as Meibomian glands (MGs). MG dysfunction is a major contributor to evaporative dry eye disease, a prevalent condition among older individuals that can lead to significant vision impairment and is associated with MG atrophy [3,4]. Each MG is composed of multiple acini that are connected to a central duct via short ductules. The central duct is lined by a stratified squamous epithelium comprising basal and suprabasal cell layers [5,6]. Within the acini, basal cells proliferate and migrate toward the center of acinus, differentiating into meibocytes that accumulate lipids as they mature. Fully differentiated meibocytes eventually disintegrate, releasing their lipid-rich content to form meibum, which flows through the ductules and central duct to the ocular surface via an opening at the eyelid margin [5,6]. Replacement of the meibocytes relies on the continual activity of stem cells (SCs) in the MG basal layer that self-renew and generate differentiating progeny [7,8]. MG dropout in aging is thought to result from reduced activity of MG SCs [7,8]. However, the precise identity of these SCs, the characteristics of their niche environment, and the regulatory mechanisms governing their function in maintaining homeostasis are only partially understood [9].

The conjunctiva is a delicate, transparent membrane that lines the inner surface of the eyelids (palpebral conjunctiva, PC) and extends to cover the sclera (bulbar conjunctiva, BC) [10]. These two regions converge at the conjunctival fornix, a site rich in goblet cells that secrete mucins such as Mucin 5AC (MUC5AC) into the tear film [11]. In addition to producing secreted mucins, conjunctival cells synthesize membrane-associated mucins, which form large oligomeric gels through the linkage of protein monomers via disulfide bonds. Secreted and membrane-associated mucins play crucial roles in maintaining the stability and functionality of the tear film [12].

The conjunctiva undergoes rapid and continuous turnover, but the exact location of the SCs responsible for its renewal is debated and may vary across species [13]. In humans, conjunctival SCs are thought to be primarily located in the medial canthal and inferior forniceal regions [14]. In contrast, in mice, SCs are believed to reside predominantly in the fornices, and are also evenly distributed throughout the BC [13,15,16]. However, specific markers that identify conjunctival SCs in mice, particularly those responsible for maintaining the PC, are lacking.

Keratin 6a (KRT6A) and Keratin 17 (KRT17) are stress keratins whose expression is associated with differentiation and inflammation [17,18]. Mutations in these keratin genes have been implicated in disorders like pachyonychia congenita, which is characterized by thickened palmoplantar skin, nail abnormalities, and white patches in mucous membranes [19,20]. KRT6A and KRT17 are expressed in hair follicles [21,22] and palmoplantar epidermis [23]. Their expression is absent from other epidermal regions in homeostasis [24,25] but is induced by cellular stress or inflammation [17,18]. Expression of *Krt6a* and *Krt17* is regulated by a variety of molecular mechanisms. These include Hedgehog (Hh) signaling, histone deacetylase 3 (HDAC3), and the transcription factor Krüppel-like factor 4 (KLF4) in various contexts [26,27,28,29].

In the MG, our recent single-nucleus transcriptomic analysis revealed that *Krt17* is expressed in the ductule, a region housing SCs that replenish MG ducts and acini [30]. Furthermore, KRT6A localizes to label-retaining cells within ductules, suggesting that it may serve as a marker for self-renewing SCs [9]. However, lineage-tracing studies that track the fates of cells expressing KRT17 or KRT6A to determine whether they contribute to adult MG homeostasis have not been performed to date, and their functions in the PC also remain unexplored.

To address these questions, we first determined the detailed expression patterns of KRT6A and KRT17 in the PC and MG using immunofluorescence (IF) and analysis of single-nucleus RNA sequencing (snRNA-seq) data. To ask whether KRT6A or KRT17 labels self-renewing SCs in the PC and MG, we carried out short- and long-term lineage-tracing assays in mice carrying *Krt6a-Cre^ERT2^* or *Krt17-Cre^ERT2^* and *Rosa26^nTnG^* or *Rosa26^mTmG^* Cre reporter alleles. To begin to elucidate the molecular mechanisms controlling expression of the *Krt6a* and *Krt17* genes, we examined the effects of inducible epithelial Hh pathway hyperactivation, *Hdac3* depletion, and *Klf4* deficiency on KRT6A and KRT17 levels in PC and MG epithelia. These experiments revealed that *Krt6a* and *Krt17* mark self-renewing SCs in the adult PC and MG, respectively. KRT17 expression is enhanced by activation of the Hh pathway, and suppressed by HDAC3 and KLF4, whereas KRT6A expression is controlled only by HDAC3.

## 2. Materials and Methods

### 2.1. snRNA-Seq Data Analysis

snRNA-seq data for MG and PC were extracted from our previously published tarsal plate tissue snRNA-seq dataset (GSE274498) [30]. The R package ‘Seurat’ (v4.3.0) was utilized for data analysis and visualization in R (v4.2.1). The signature genes used by Seurat to define each cell population are listed in Appendix A and Table A1.

### 2.2. Mice

The following mouse lines were used: *Krt6a-Cre^ERT2^* (Strain #035093, The Jackson Laboratory, Bar Harbor, ME, USA) [21]; *Krt17-Cre^ERT2^* [31]; *Rosa26^nTnG^* (Strain #023537, The Jackson Laboratory); *Rosa26^mTmG^* (Strain #007676, The Jackson Laboratory); *Hdac3^fl^* [32]; *Krt5-rtTA* [33]; *tetO-Cre* [34]; *tetO-GLI2ΔN* [35]; and *Klf4^fl^* [36]. Mice were assigned to experimental or control groups based on their genotypes, and inclusion in the analysis was determined accordingly. Investigators were aware of the genotypes during the allocation and handling of the mice, as this information was essential for proper treatment and management. Mice were housed in groups of up to five per cage in a specific pathogen-free barrier facility on a 12-h light/12-h dark cycle at 65–75 °F with 40–60% humidity and fed with standard rodent laboratory chow (5001, Purina, Gray Summit, MO, USA). DNA recombination was induced by administering doxycycline chow (6 g/kg, Bio-Serv, Flemington, NJ, USA) or 200 µg/mL doxycycline (D9891, Sigma-Aldrich, Saint Louis, MO, USA) dissolved in 5% sucrose water for the specified duration. For lineage tracing, each mouse was given a single intraperitoneal injection of 200 μL Tamoxifen (T5648, Sigma-Aldrich) dissolved in corn oil (10 mg/mL). All mice were maintained on a mixed-strain background. All animal experiments were conducted under approved protocols (approval number: IACUC-2019-0028) according to the institutional guidelines of the Icahn School of Medicine at Mount Sinai IACUC committee, following the guidelines within the ARVO statement for the use of animals in ophthalmic and vision research. At least three mice from each genotype and experimental condition were included in the assays, with both male and female mice represented in the study. Mouse euthanasia was performed by exposing the animals to CO_2_ for 5 min, followed by cervical dislocation. The exact number of mice analyzed for each assay is provided in the corresponding figure legend.

### 2.3. Human Eyelid Samples

Human eyelid tissues containing normal MG or Meibomian gland carcinoma (MGC) tissues were obtained from the Department of Ophthalmology, University of Pennsylvania. The samples were de-identified specimens discarded from unrelated surgical procedures, and informed consent from patients was waived. Use of these tissues for research was approved by the University of Pennsylvania IRB (protocol 827926, PI Vivian Lee). Samples were paraffin-embedded and sectioned for IF analysis. The number of human eyelid samples used for histological analysis is specified in the figure legend.

### 2.4. Bromodeoxyuridine (BrdU) Pulse-Chase Assay

Wild-type 8-week-old *C57BL/6J* mice (Strain #000664, The Jackson Laboratory) were injected with 200 µL of BrdU/PBS solution (10 mg/mL). Mice were euthanized at the designated time points, and their eyelids were collected for fixation and subsequent histological analysis. Two males and two female mice were analyzed at each time point.

### 2.5. Histology and IF

Tissues were fixed overnight at 4 °C in 4% paraformaldehyde (PFA)/PBS (Affymetrix/USB, Santa Clara, CA, USA), then embedded in paraffin and sectioned at 5 µm using a Leica HistoCore BIOCUT microtome (Leica, Wetzlar, Germany). The sections were de-paraffinized and rehydrated using xylene substitute (Sigma-Aldrich) and graded ethanol solutions. Antigen retrieval was performed using the unmask solution (Vector Labs, Newark, CA, USA). For IF, sections were incubated with primary antibodies overnight at 4 °C, followed by incubation with dye-conjugated secondary antibodies (Invitrogen and Vector Labs) and mounting in a DAPI-containing medium (Invitrogen). Whole-mount images of the tarsal plate were captured using a Leica SP8 laser scanning confocal microscope equipped with Power HyD detectors. The images were processed with Leica LAS X software (v3.6.0.20104) for stacked image maximum projection and 3D image reconstruction.

The following primary antibodies were used: rabbit anti-HDAC3 (Santa Cruz, Dallas, Texas, USA, sc-11417, 1:100); Guinea pig anti-PLIN2 (Fitzgerald, Acton, MA, USA, 20R-AP002, 1:500); rabbit anti-GFP (Abcam, Cambridge, UK, ab290, 1:1000), rabbit anti-KRT6A (BioLegend, San Diego, California, USA, 905702, 1:200); mouse anti-BrdU (Cell signaling, Danvers, MA, USA, 5292S, 1:200), mouse anti-MYC-tag (Cell Signaling, 2276S, 1:1000); goat anti-KLF4 (R&D Systems, AF3158, 1:100); chicken anti-KRT14 (BioLegend, 906004, 1:500); rat anti-Ki-67 (eBioscience, San Diego, California, USA, 14-5698-82, 1:200); rabbit anti-KRT17 (Abcam, ab109725, 1:200).

The following secondary antibodies were used: donkey anti-rabbit IgG, Alexa Fluor 555 (Invitrogen, Waltham, MA, USA, A-31572, 1:800); donkey anti-goat IgG Alexa Fluor 488 (Invitrogen, A-11055, 1:800), donkey anti-mouse IgG, Alexa Fluor 488 (Invitrogen, A-21202, 1:800); donkey anti-chicken IgG Alexa Fluor 647 (Invitrogen, A78952, 1:800); goat anti-guinea pig, Biotinylated (Vector Labs, BA-7000, 1:500); Streptavidin Fluorescein (Vector Labs, SA-5001, 1:500); Streptavidin Texas Red (Vector Labs, SA-5006, 1:500).

### 2.6. RNAscope

RNAscope was performed on paraffin sections according to the protocol provided by Advanced Cell Diagnostics (ACD, Newark, CA, USA), using the RNAscope Multiplex Fluorescent Reagent Kit v2 (ACD #323100) and probes targeting *Ptch1* (ACD #402811). The sections were examined and photographed with a Leica Microsystems DM5500B fluorescent microscope.

## 3. Results

### 3.1. KRT6A and KRT17 Are Differentially Expressed in Adult PC and MG

To investigate the expression of KRT6A and KRT17 in the PC and MG, we conducted IF analysis on adult mouse eyelid tissues. The results showed that KRT6A is strongly expressed in the PC, mainly in the suprabasal layer (white arrows, Figure 1A,B). Moderate expression of KRT6A was also observed in some Ki-67+ proliferating basal cells of the PC (yellow arrow, Figure 1B). In the MG, KRT6A was highly expressed in suprabasal cells of the central duct (green arrows, Figure 1B,C) and moderately expressed in the ductule (blue arrow, Figure 1C), as well as in a few acinar basal cells (red arrow, Figure 1C). By contrast, KRT17 was not detected in the PC epithelium (pink arrow, Figure 1D,E), but showed robust expression in suprabasal ductal cells of the MG (green arrows, Figure 1D–F). KRT17 expression was also observed in the MG ductule (blue arrow, Figure 1F) and in some Ki-67+ ductal basal cells (yellow arrow, Figure 1E). Additionally, both KRT6A and KRT17 were expressed in the eyelid hair follicles (orange arrows, Figure 1A,D), in agreement with previous reports [37]. Corresponding with the IF findings, our snRNA-seq analysis confirmed that *Krt6a*, but not *Krt17*, shows broad and strong expression in PC cells. In the MG, *Krt6a* and *Krt17* exhibit partially overlapping expression, primarily in the suprabasal ductal cells (Figure 1G). Together, these findings highlight the distinct expression patterns of KRT6A and KRT17 in adult PC and MG epithelium.

### 3.2. Krt6a-Expressing Cells Contribute to Adult PC Homeostasis

To determine whether *Krt6a*-expressing cells mark self-renewing SCs in adult PC and MG, we first assessed the approximate turnover time of the adult PC epithelium using BrdU pulse-chase analysis (Figure 2A). This assay labels proliferative cells as they incorporate BrdU during DNA synthesis. Over time, the BrdU label is gradually diluted and eventually becomes undetectable after multiple rounds of cell division [38]. Two hours after BrdU injection, we observed BrdU+ basal cells in the PC epithelium (white arrows, Figure 2B). These BrdU+ basal cells persisted through 2-day and 7-day chase periods (white arrows, Figure 2C,D) and produced progeny in the suprabasal layer (yellow arrows, Figure 2C,D). After a 14-day chase, BrdU+ cells were no longer detectable in the PC epithelium (Figure 2E), indicating that the turnover of most PC cells occurs within 14 days. Based on these data, lineage-tracing assays conducted over several months would be sufficient to identify self-renewing SCs in the adult PC and MG epithelium [30].

To determine whether *Krt6a-Cre^ERT2^* activity mirrors the endogenous expression of KRT6A in adult PC and MG epithelia, we performed short-term lineage tracing in *Krt6a-Cre^ERT2^ Rosa26^nTnG^* mice (Figure 3A). After 48 h of tracing, whole-mount confocal imaging revealed widespread GFP+ cells throughout the PC epithelium (Figure 3B and Appendix A). GFP+ cells were also observed in the MG central duct (yellow arrow, Figure 3C), MG ductule (blue arrow, Figure 3C), and hair follicles (yellow arrow, Figure 3C). Further IF analysis confirmed that most Krt6a^GFP^+ cells were present in the PC suprabasal layer (yellow arrow, Figure 3D), with a few located in the basal layer of the PC (pink arrow, Figure 3D) and in the suprabasal layer of the MG central duct (white arrow, Figure 3D). Thus, *Krt6a-Cre^ERT2^* is active in the same cell populations that express KRT6A protein (Figure 1A–D) and mRNA (Figure 1G) in adult PC and MG epithelia.

To ask whether *Krt6a*-expressing cells self-renew and give rise to differentiating cells in the PC and MG, we analyzed *Krt6a-Cre^ERT2^ Rosa26^nTnG^* mice 3 months after inducing Cre activity. These assays identified multiple clones of Krt6a^GFP^+ cells within the PC (blue arrows, Figure 3E; yellow arrow, Figure 3G). GFP+ cells were also present in the hair follicles (white arrow, Figure 3F). However, we did not detect any GFP+ cells in the MGs (Figure 3F and Appendix A). Together, these results demonstrate that in the PC, KRT6A marks SCs that self-renew and contribute to adult homeostasis. By contrast, KRT6A does not mark SCs in the adult MG.

### 3.3. KRT17 Marks Self-Renewing SCs That Maintain Adult MG

To evaluate whether KRT17-expressing cells are involved in the maintenance of adult PC and MG, we performed lineage-tracing assays using the *Krt17-Cre^ERT2^ Rosa26^mTmG^* mice (Figure 4A). Short-term lineage-tracing results showed that *Krt17-Cre^ERT2^* did not label any PC cells (blue arrow, Figure 4B). In the MG, *Krt17-Cre^ERT2^* exhibited activity in the suprabasal layer of the duct (red arrows, Figure 4B,C), basal layer of the duct (white arrow, Figure 4C), ductular cells (yellow arrow, Figure 4C), and basal layer of the acinus (pink arrow, Figure 4C). These findings indicate that *Krt17-Cre^ERT2^* activity recapitulates the endogenous expression pattern of KRT17 protein (Figure 1D–F) and mRNA (Figure 1G).

To ask whether KRT17 marks self-renewing MG SCs that give rise to differentiating progeny, we analyzed *Krt17-Cre^ERT2^ Rosa26^mTmG^* 3 months after inducing Cre activity. This analysis revealed that KRT17-expressing cells generated progeny that contributed to the maintenance of both MG duct (white arrows, Figure 4D,E) and acinus (pink arrows, Figure 4D,E). These observations indicate that KRT17 is expressed in self-renewing SCs in the adult MG but not in the PC.

### 3.4. Forced Activation of the Hh Signaling Pathway Induces KRT17, but Not KRT6A Expression, in Adult MG and PC Epithelia, and KRT17 Is Broadly Expressed in Human MGC Samples

To begin to investigate the regulatory mechanisms underlying the expression of KRT6A and KRT17 in the PC and MG, we initially focused on the Hh signaling pathway, as both genes can be activated by Hh signaling in epidermal keratinocytes. To determine whether forced activation of Hh signaling elevates expression of KRT6A or KRT17 in the PC and/or MG, we examined their expression in *Krt5-rtTA tetO-GLI2ΔN* mice that can be induced to express an activated form of the Hh pathway activator GLI2 (GLI2ΔN) in epithelial cells by treatment with doxycycline [39] (Figure 5A). After 2 days of doxycycline treatment, RNAscope assays showed a significant increase in expression of the Hh pathway target gene *Ptch1* in GLI2ΔN-overexpressing PC and MG tissues (Figure 5B,C). IF analysis revealed that overexpression of GLI2ΔN induced ectopic expression of KRT17 in the PC (green arrow, Figure 5D,E) and upregulated KRT17 expression in the acinar basal layer of the MG (yellow arrows, Figure 5D,E). However, the expression of KRT6A remained unchanged in GLI2ΔN-overexpressing PC and MG epithelium compared to control tissues (Figure 5F,G).

Given that human MGC is characterized by expansion of GLI2+ cells concomitant with expression of several MG SC markers [30], we next examined KRT6A and KRT17 expression in human MG and MGC tissues. Consistent with previous findings [40], IF staining showed that both KRT6A and KRT17 were primarily localized to the MG duct (yellow arrows, Figure 5H,J) and ductule (white arrows, Figure 5H,J). Notably, KRT17 expression was readily detected in MGC tissues, whereas KRT6A expression was undetectable (Figure 5I,K). Therefore, hyperactivation of the Hh pathway specifically induces KRT17 expression, but does not affect KRT6A expression, in adult PC and MG, and expansion of GLI2+ undifferentiated cells in human MGC is associated with broad KRT17 expression without detectable KRT6A expression.

### 3.5. HDAC3 Suppresses KRT17 and KRT6A Expression in Adult PC and MG Acini

We showed previously that KRT17 expression is suppressed by the histone deacetylase HDAC3 in MG acinar cells [29] (Appendix A, yellow arrow, Appendix A). To determine whether similar mechanisms control KRT17 expression in the PC, and KRT6A expression in MG and PC, we generated *Krt5-rtTA tetO-Cre Hdac3^fl/fl^* (*Hdac3^cKO^*) mice in which deletion of epithelial *Hdac3* can be induced by doxycycline. Experimental and control mice were doxycycline treated starting at P48 and expression of HDAC3, KRT17 and KRT6A in PC epithelium was analyzed after 10 days (Figure 6A). These analyses revealed efficient deletion of HDAC3 in adult PC (Figure 6B,C) and strong induction of KRT17 expression in *Hdac3*-deficient PC (Figure 6D, white arrow, Figure 6E). Similarly, HDAC3 deletion caused increased expression of KRT6A in the MG acinus (Appendix A, yellow arrow, Appendix A) and in the PC basal layer (Figure 6F, white arrows, Figure 6G). As upregulated Hh signaling activity is observed in *Hdac3*-deficient MG acini [29], we assessed *Ptch1* expression in the PC. RNAscope assay data showed that *Hdac3* deletion also resulted in elevated Hh activity in PC (Figure 6H, white arrows, Figure 6I). These findings suggest that HDAC3 represses KRT17 expression in the MG acinus and PC basal layer, possibly through restricting Hh pathway activity. Additionally, HDAC3 inhibits KRT6A expression in these tissues; as Hh signaling does not appear to affect KRT6A expression, the underlying mechanism for this inhibition requires further investigation.

### 3.6. KLF4 Represses KRT17 Expression in Adult PC

The transcription factor KLF4 contributes to the regulation of KRT17 expression in embryonic and adult interfollicular epidermis [28] (Appendix A). To test whether KLF4 also influences KRT17 expression in the MG and PC, we generated *Krt5-rtTA tetO-Cre Klf4^fl/fl^* (*Klf4^cKO^*) mice in which deletion of epithelial *Klf4* can be induced by doxycycline. Control and *Klf4^ckO^* mice were placed on oral doxycycline starting at P30 (Figure 7A). IF analysis of KLF4 and KRT17 expression at P61 revealed that KLF4 is primarily expressed in PC suprabasal cells (yellow arrow, Figure 7B) and MG duct suprabasal cells (pink arrow, Figure 7B), but is absent from the MG acinus (green arrow, Figure 7B). KLF4 was efficiently deleted from both the PC and MG following doxycycline administration (Figure 7C).

KRT17 was ectopically expressed in *Klf4*-deficient PC (Figure 7D, white arrow, Figure 7E), but showed no significant increase relative to controls in the MG (Appendix A). By contrast, while *Klf4* deletion led to ectopic expression of KRT6A in the eyelid interfollicular epidermis (Appendix A), neither KRT6A nor *Ptch1* expression in the PC or MG was affected in *Klf4*-deficient mutants (Figure 7F–I and Appendix A). Collectively, these findings suggest that KLF4 plays a crucial role in repressing KRT17 expression within the adult PC epithelium, but does not influence KRT6A or Hh signaling activity in the PC and MG.

## 4. Discussion

Both KRT6A and KRT17 are expressed in normal human and mouse MG, and their expression and functions in other ocular tissues have been described. For example, KRT6A expression is upregulated in pterygium, a benign overgrowth of BC [41], while KRT17 marks limbal SCs and regulates their stemness, differentiation, proliferation, and immune responses [42]. However, their expression patterns in the PC and their functional roles in MG and PC remain elusive. In this study, we show that KRT6A and KRT17 are differentially expressed in the adult mouse MG and PC, provide direct evidence that they mark distinct SC populations in these tissues, and elucidate their regulation by Hh signaling, HDAC3 and KLF4 in these tissues. Nevertheless, due to limited access to normal human PC, the expression of KRT6A and KRT17 in this tissue remains to be determined.

While SCs responsible for replenishing the conjunctiva have long been recognized, specific markers for these SCs are still poorly characterized. Recent research identified Nerve Growth Factor Receptor (NGFR) as a marker for bipotent conjunctival SCs capable of generating goblet cells and keratinocytes within human conjunctival organoids [43]. However, few in vivo lineage-tracing studies have been performed. Using *Krt6a-Cre^ERT2^ Rosa26^nTnG^* mice, we provide compelling evidence that KRT6A marks SCs involved in PC homeostasis. Notably, we did not observe stripes of Krt6a^GFP^+ cells emanating from the corneal limbus, as would be expected if the limbus was the source of KRT6A+ SCs [44]; instead, KRT6A+ SCs in the basal PC appear to locally replenish adjacent PC cells. In line with this, lineage tracing with *KRT14-Cre^ERT2^ Rosa26^TdTomato^* mice revealed no movement of TdTomato+ clones from the cornea to the PC, indicating the existence of distinct SC populations for PC and cornea, each maintaining their respective compartments [45].

Similarly, the identity of SCs maintaining adult MG is only partially understood. Previous findings suggested that MG ducts and acini are maintained by distinct unipotent KRT14+ SCs [9]. However, KRT14 is ubiquitously expressed in MG [29], and specific markers for these subpopulations were not well defined. Recent lineage-tracing data indicate that *Slc1a3*+ basal SCs contribute exclusively to acini, while *Lrig1*+, *Lgr6*+, *Axin2*+, or *Gli2*+ SCs replenish both ducts and acini [30]. Our current lineage-tracing data demonstrate that KRT17 also marks SCs involved in MG homeostasis.

Interestingly, despite the finding that KRT6A is expressed in label-retaining ductular cells [9], in our lineage-tracing studies, KRT6A-expressing cells failed to mark SCs in the adult MG. It is unlikely that this discrepancy was due to incomplete labeling efficiency of KRT6A+ cells in the MGs of *Krt6a-Cre^ERT2^ Rosa26^nTnG^* mice, as short-term lineage tracing confirmed that labeled cells corresponded to endogenous KRT6A expression in the MG duct suprabasal layer and ductular cells. Furthermore, long-term lineage tracing identified clones of GFP+ cells in the PC and hair follicles, indicating that *Krt6a-Cre^ERT2^ Rosa26^nTnG^* marked SCs in other tissues. One potential explanation is that, for detection of label-retaining cells in the earlier study, the cells were initially labelled during MG development, while our study focused on adult MG cells. Alternatively, the tet-off system used [9] may not have completely suppressed H2B-GFP expression in all MG cells, allowing a few KRT6A+ ductular cells to retain H2B-GFP. Further investigation is necessary to explore these possibilities.

Overexpression of KRT6A and KRT17 has been linked to hyperproliferation and the progression of various cancers, prompting extensive research into the mechanisms regulating their expression [37]. In particular, both *Krt6a* and *Krt17* promoters can be directly activated by GLI2 overexpression [26]. In line with this, we found that hyperactivation of the Hh signaling pathway, driven by forced expression of GLI2ΔN, was sufficient to induce ectopic KRT17 expression in the PC and in MG acini. Interestingly, KRT6A expression was unaffected by GLI2ΔN overexpression in this system. This may be because its promoter exhibits lower sensitivity to GLI2 activation compared to the *Krt17* promoter [26]. In line with this observation, KRT17, but not KRT6A, was broadly expressed in human MGC tissues, which originate from GLI2+ proliferative, poorly differentiated cells that also express MG stem cell markers. This finding suggests that KRT17 expression is also associated with hyperproliferation in MGC and further supports our theory that KRT17+ cells, but not KRT6A+ cells, serve as SCs contributing to MG homeostasis.

HDAC3 deletion led to increased Hh pathway activity in the PC and MG, potentially contributing to the ectopic expression of KRT17 in *Hdac3*-deficient PC and MG acini. However, the precise mechanism by which HDAC3 suppresses Hh pathway activity in these tissues remains unclear. Alternatively, HDAC3 may cooperate with KLF4 to repress KRT17 expression in the PC, as observed in the developing interfollicular epidermis [28]. KLF4 is not expressed in MG acini, indicating that HDAC3′s inhibitory effect on KRT17 expression in these cells is likely independent of KLF4.

Our current and previous data showed that KLF4 deletion causes ectopic expression of KRT6A in adult and embryonic interfollicular epidermis [28]. Furthermore, deletion of KLF4 in embryonic ocular surface results in severe abnormalities, including defective corneal development, loss of conjunctival goblet cells, and hyperproliferative epithelial cells [46]. By contrast, we found that inducible deletion of KLF4 in adult ocular epithelia caused ectopic expression of KRT17 in PC but not MG and did not alter KRT6A expression in either tissue. Therefore, regulation of KRT6A by KLF4 appears to be context dependent. Further studies will be needed to determine whether HDAC3 and KLF4 regulate the expression of KRT6A and KRT17 by directly binding their promoter or enhancer regions in ocular surface cells, or through indirect mechanisms.

Despite these findings, our study has several limitations. First, we did not examine KRT6A and KRT17 expression in human PC tissues because normal PC was not available in the eyelid samples we analyzed. Assessing their expression in human specimens would help determine whether their expression patterns are conserved across species. Second, it remains unclear whether clinical variables, such as age, sex, or cancer stage, affect KRT6A and KRT17 expression in human MG and MGC, which should be evaluated in a larger cohort. Third, we did not investigate the functional roles of these stress keratins in PC and MG homeostasis or disease. Future studies using genetically modified mouse models to manipulate their expression could provide important mechanistic insights. Finally, the functional roles of HDAC3 and KLF4 in adult PC homeostasis require further characterization.

In summary, our findings reveal distinct contributions of KRT6A- and KRT17-expressing SCs in adult PC and MG homeostasis. Additionally, we identify key regulators that control the expression of KRT6A and KRT17 in these tissues. Our study offers new insights into SC biology within the ocular surface, enhancing our understanding of the mechanisms that sustain these critical tissues.

## Figures and Tables

**Figure 1 cells-14-01979-f001:**
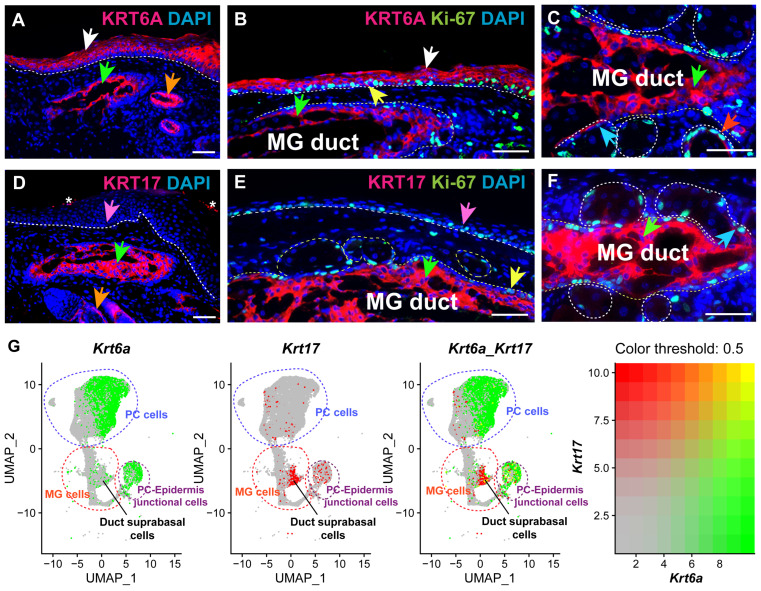
KRT6A and KRT17 are differentially expressed in adult PC and MG epithelia. (**A**) IF showed robust expression of KRT6A within adult mouse PC epithelium (white arrow), MG duct (green arrow), and hair follicles (orange arrow), while it was absent in eyelid interfollicular epidermis. The white dashed line indicates the boundary between epithelium and mesenchyme; the green dashed line indicates the interface between PC and interfollicular epidermis. (**B**) IF indicated strong expression of KRT6A in PC suprabasal cells (white arrow) and MG duct suprabasal cells (green arrow), and moderate expression in some Ki-67+ PC basal cells (yellow arrow). The white dashed lines outline the PC epithelium and MG duct. (**C**) Higher magnification view of IF staining shows strong expression of KRT6A in MG duct suprabasal cells (green arrow) and weak KRT6A expression in MG ductule (blue arrow) and acinar basal cells (red arrow). The white dashed lines outline the MG duct and acini. (**D**) IF indicates KRT17 expression predominantly in MG duct (green arrow) and hair follicles (orange arrow), and absence in PC epithelium (pink arrow). The white dashed line outlines the PC epithelium. The white asterisks indicate background noise signals. (**E**) IF demonstrates robust KRT17 expression in MG duct suprabasal cells (green arrow) and Ki-67+ duct basal cells (yellow arrow), while it is not expressed in PC epithelium (pink arrow). The white dashed lines outline the PC epithelium, MG duct, and acini. (**F**) Higher magnification view of IF staining reveals robust expression of KRT17 in MG suprabasal cells (green arrow) and weak expression in Ki-67+ MG ductular cells (blue arrow). (**G**) Feature plots from snRNA-seq data analysis showing the expression of *Krt6a* (green) and *Krt17* (red) in PC and MG cell populations. The yellow color indicates cells strongly expressing both *Krt6a* and *Krt17*. *N * =  3 mice were analyzed for IF staining. Scale bars: 50 μm.

**Figure 2 cells-14-01979-f002:**
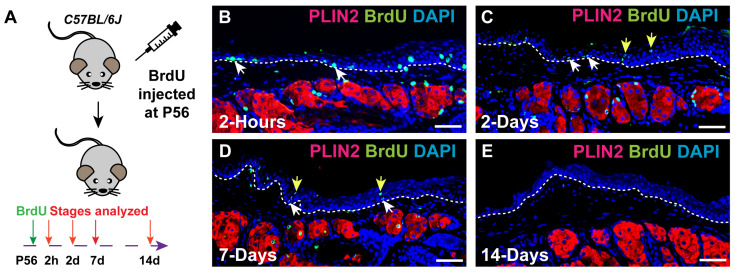
Adult PC epithelial cells turn over within 14 days. (**A**) Schematic illustrating the experimental setup. (**B**) IF reveals BrdU+ proliferative basal cells within adult PC epithelium (white arrows) at 2 h after BrdU administration. (**C**) IF reveals BrdU+ basal cells (white arrow) and suprabasal basal cells (yellow arrow) within PC epithelium at 2 days after BrdU injection. (**D**) IF demonstrated the presence of BrdU+ basal cells (white arrows) and suprabasal cells (yellow arrows) within PC epithelium at 7 days after BrdU administration. (**E**) IF confirms the absence of BrdU+ cells within PC epithelium in mice at 14 days after BrdU administration (14-day chase). PLIN2 marks meibocytes. White dashed lines delineate the PC. *N* = 4 mice were analyzed at each time point. Scale bars: 50 μm.

**Figure 3 cells-14-01979-f003:**
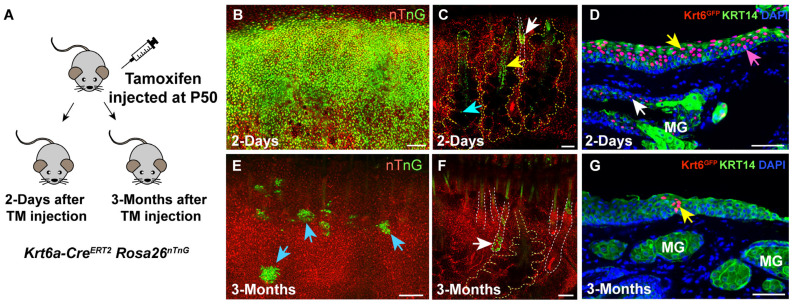
KRT6A marks self-renewing SCs in PC, but not in MG. (**A**) Schematic depicting the breeding strategy and experimental setup. (**B**,**C**) Whole-mount fluorescent images showing the presence of Krt6a^GFP^+ cells in PC (**B**), MG ductule ((**C**), blue arrow), MG duct ((**C**), yellow arrow), and hair follicle ((**C**), white arrow) in mice lineage traced for 2 days. (**D**) IF results demonstrating the presence of Krt6a^GFP^+ cells in PC suprabasal layer (yellow arrow), PC basal layer (pink arrow), and MG duct suprabasal layer (white arrow) in mice following 2 days of lineage tracing. (**E**,**F**) Whole-mount fluorescent images indicating the presence of Krt6a^GFP^+ cell clones in PC ((**E**), blue arrows) and hair follicle ((**F**), white arrow) in mice lineage traced for 3 months. (**G**) IF results illustrating a Krt6a^GFP^+ cell clone within PC epithelium (yellow arrow) in mice following 3 months of lineage tracing. Yellow and white dashed lines delineate the MGs and hair follicles, respectively. *N* = 4 mice were analyzed at each time point. For whole-mount imaging, entire mouse eyelids were scanned to ensure complete detection of GFP+ cells within the MG and to avoid potential omission. Scale bars in panels (**B**,**C**,**E**,**F**) represent 100 µm; scale bars in panels (**D**,**G**) indicate 50 µm.

**Figure 4 cells-14-01979-f004:**
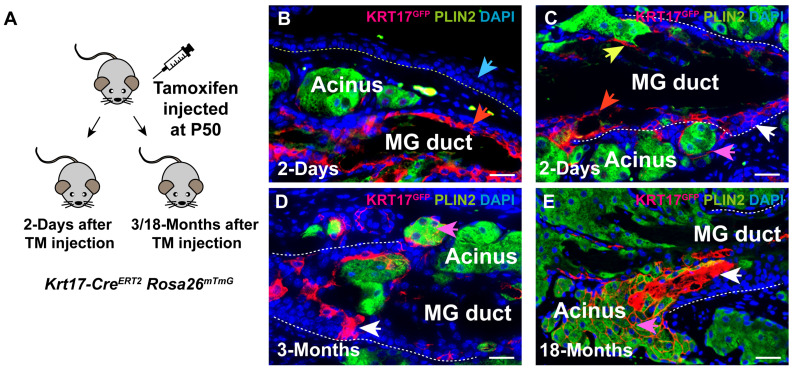
KRT17-expressing cells contribute to adult MG homeostasis. (**A**) Schematic depicting the breeding strategy and experimental setup. (**B**,**C**) IF results demonstrate that Krt17^GFP^+ cells are absent in PC epithelium (blue arrow, (**B**), but are present in MG duct suprabasal basal layer (red arrow, (**B**,**C**), duct basal layer (white arrow, (**C**), acinar basal layer (pink arrow, (**C**), and ductule (yellow arrow, (**C**) in mice following 2 days of lineage tracing. (**D**,**E**) IF identifies Krt17^GFP^+ cells in MG duct (white arrows in (**D**,**E**) and acini (pink arrows in (**D**,**E**) in mice following 3 months (**D**) or 18 months (**E**) of lineage tracing. The yellow dashed line in (**B**) delineates PC, while the white dashed lines in (**C**–**E**) delineate the MG duct. KRT14 marks epithelial cells. *N* = 3 mice were assessed at each time point. Scale bars: 25 μm.

**Figure 5 cells-14-01979-f005:**
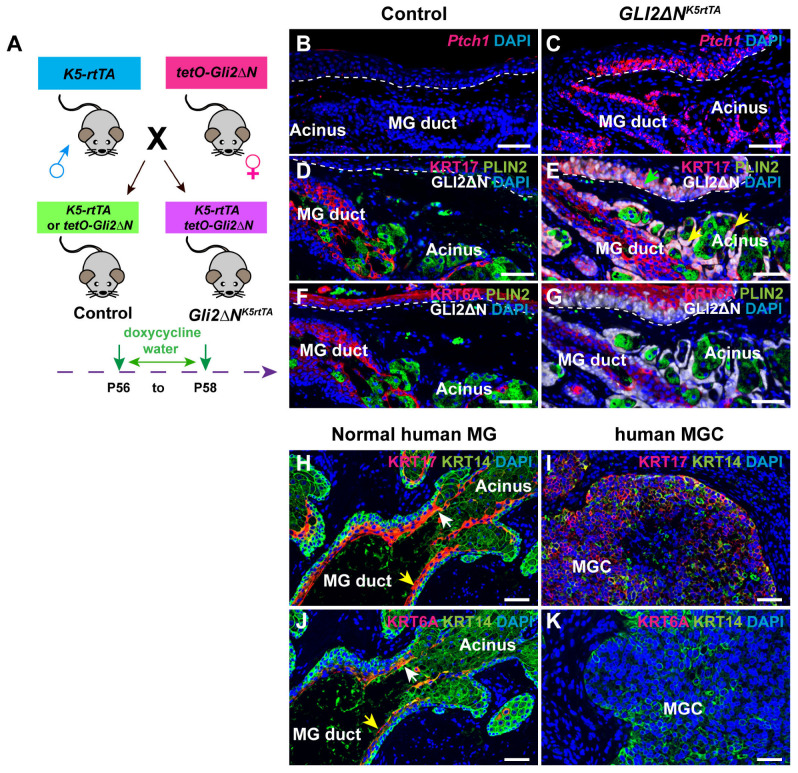
Forced activation of *Hh* signaling induces KRT17 expression in adult PC and MG, and human MGC is characterized by broad KRT17 expression. (**A**) Schematic depicting the breeding strategy and experimental setup. (**B**,**C**) RNAscope data showing expression of *Ptch1* in the PC and MG of control (**B**) and GLI2ΔN-overexpressing (**C**) mice subjected to 2 days of doxycycline treatment. (**D**,**E**) IF data indicates that GLI2ΔN induces robust KRT17 expression in PC ((**E**), green arrow) and acinar basal layer ((**E**), yellow arrow), whereas KRT17 is primarily expressed in MG duct (**D**). (**F**,**G**) IF reveals that expression of KRT6A is similar between control (**F**) and GLI2ΔN-overexpressing (**G**) PC and MG cells. White dashed lines delineate the PG. (**H**,**J**) IF results demonstrate that KRT17 and KRT6A are primarily localized to the MG duct (yellow arrows) and ductule (white arrows) in normal human MGs. (**I**,**K**) IF staining revealing broad KRT17 expression (**I**) and absence of detectable KRT6A expression (**K**) in human MGC samples. *N* = 4 mice of each genotype, *N* = 3 normal human eyelid samples, and *N* = 5 human MGC samples were analyzed. Scale bars: 50 μm.

**Figure 6 cells-14-01979-f006:**
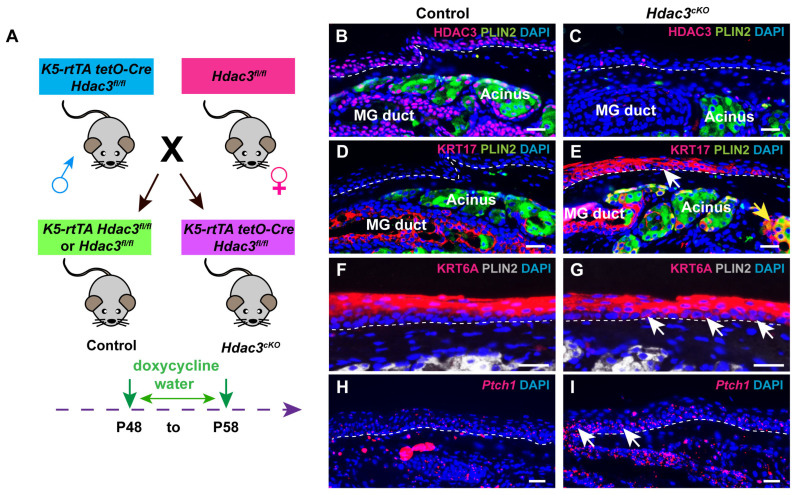
HDAC3 negatively regulates KRT6A and KRT17 expression in adult PC and MG. (**A**) Schematic depicting the breeding strategy and experimental setup. (**B**,**C**) IF demonstrates that HDAC3 is broadly expressed in control PC and MG (**B**) and is effectively deleted in the PC and MG of *HDAC3^cKO^* mice at 10 days after initiating doxycycline treatment (**C**). (**D**) IF reveals absence of KRT17 expression in PC and positive KRT17 staining in the MG duct. (**E**) IF shows that *Hdac3* deletion causes ectopic KRT17 expression in PC (white arrow) and MG acinus (yellow arrow). (**F**) IF shows that *Hdac3* deletion results in increased KRT6A expression in the PC suprabasal layer. (**G**) IF reveals increased expression of KRT6A in the basal layer of PC following *Hdac3* deletion (white arrows). (**H**,**I**) RNAscope shows weak *Ptch1* expression within the control PC epithelium (**H**) and increased *Ptch1* expression in *Hdac3*-deficient PC epithelium ((**I**), white arrows). White dashed lines delineate the PC. *N* = 4 mice per genotype were analyzed. Scale bars: 25 μm.

**Figure 7 cells-14-01979-f007:**
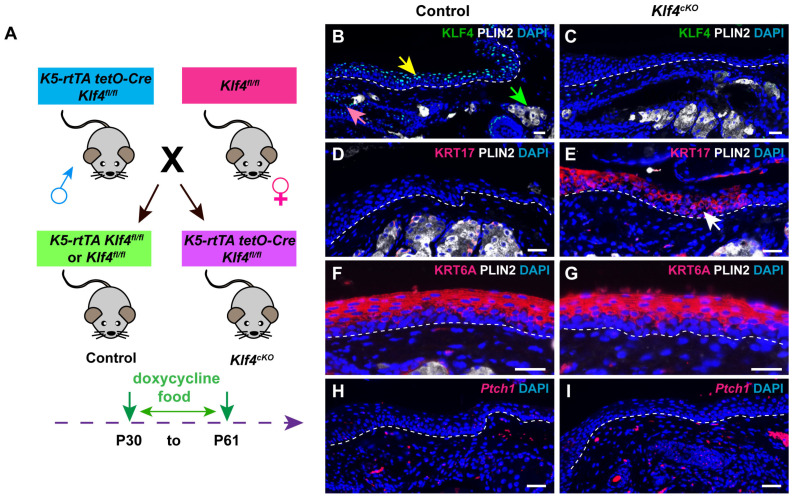
*Klf4* deletion causes ectopic expression of KRT17 in adult PC epithelium. (**A**) Schematic depicting the breeding strategy and experimental setup. (**B**) IF shows robust KLF4 expression in the suprabasal layer of PC (yellow arrow) and MG duct (pink arrow) but no KLF4 expression in the MG acinus (green arrow) of control mice. (**C**) IF confirms efficient deletion of KLF4 in the PC and MG of *Klf4^cKO^* mice at 31 days after initiating doxycycline treatment. (**D**) IF demonstrates the absence of KRT17 in control PC. (**E**) IF reveals ectopic expression of KRT17 in *Klf4*-deficient PC (white arrow). (**F**,**G**) IF shows that expression of KRT6A is comparable between control (**F**) and *Klf4*-deficient (**G**) PC. (**H**,**I**) RNAscope reveals similar levels of *Ptch1* mRNA expression in control (**H**) and *Klf4*-deficient (**I**) PC. White dashed lines delineate the PC. *N* = 3 mice per genotype were analyzed. Scale bars: 25 μm.

## Data Availability

The authors declare that the main data supporting the findings of this study are available within the article and its Appendix A. All correspondence and material requests should be addressed to Xuming Zhu or Sarah E. Millar. The snRNA-seq dataset can be accessed from the GEO database (https://www.ncbi.nlm.nih.gov/geo/) under accession number GSE274498.

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
