# Peer review of "KRT6A and KRT17 Mark Distinct Stem Cell Populations in the Adult Palpebral Conjunctiva and Meibomian Gland"

_cells, 2025, doi:10.3390/cells14241979_

Round 1
Reviewer 1 Report
Comments and Suggestions for Authors
In this interesting report Zhu et al. aim to understand if two stress-related keratins, KRT6A and KRT17, can mark self-renewing stem cells in adult mouse Meibomian gland and palpebral conjunctiva. The authors looked at where these keratins are expressed using single-nucleus RNA sequencing and immunofluorescence. They also employed lineage tracing to see if the cells expressing these keratins keep renewing over time. They found that both KRT6A and KRT17 appear mainly in the central duct and the ductules of the Meibomian gland, while KRT6A is also strongly present in the conjunctiva. The tracing experiments showed that Krt17 marks the stem cells that maintain the Meibomian gland, and Krt6a marks the ones that take care of the conjunctiva. Then they tested some molecular regulators to see how they change keratin expression. When they overexpressed GLI2ΔN, a Hedgehog pathway activator, the authors observed ectopic expression of KRT17 but not for KRT6A. Removing Hdac3 caused both keratins to expand their expression in the gland and also KRT17 in the conjunctiva. Removing Klf4 mainly caused abnormal KRT17 in the conjunctiva but didn’t change KRT6A. Overall in this study the authors found that Krt6a and Krt17 expressing stem cells are mantaining the conjunctiva and the Meibomian gland respectively. In addition they provide mechanistic insights that regulate these two keratins.
The manuscript unfold logically and contain interesting results. The figures are well presented. Here some points to take in consideration before publications:
As meibomian glands MG are specialized sebaceous glands SG, if the authors have collected also skin samples, it would be interesting to check if some of the conclusions made on the MG are also treu for the SG in the skin.
The analysis in Figure 1 G does not provide information of previously published markers of MG and PC compartments. For examples Gata6 for the SG ducts.
Although the scRNA-seq was already published there are not yet available text tables with markers of each cluster/cell types. In this manuscript the authors should provide a Table with differentially expressed gene per cluster/celltypes in the supplemetary data. This would represent a valuable resource for the scientist interested in Meibomian gland and palpebral conjunctiva.
While le manuscript is rich of interesting histological observations there is a general lack of quantification.
Are goblet cells in the conjuntiva also progenies of Krt6a cells?
A schematic at the end of the manuscript, summarising the results could help the readers.
Author Response
As meibomian glands MG are specialized sebaceous glands SG, if the authors have collected also skin samples, it would be interesting to check if some of the conclusions made on the MG are also treu for the SG in the skin.
Response: We appreciate the reviewer’s valuable advice. The role of KRT6A in sebaceous glands (SGs) has been described previously (see ref. 26), so we did not specifically focus on this aspect. However, we did examine whether Krt17⁺ progeny contribute to hair follicle–associated SGs in the eyelid skin. Our lineage tracing experiments showed that KRT17+ lineages replenish both SGs and hair follicles. Because Krt17 is broadly expressed in the hair follicle and SG, and SG homeostasis is maintained by multiple compartments, it is challenging to determine which specific Krt17⁺ cell population contributes to SG maintenance; therefore, we did not pursue this further.
The analysis in Figure 1 G does not provide information of previously published markers of MG and PC compartments. For examples Gata6 for the SG ducts.
Response: The markers used to identify the different clusters in our snRNA-seq dataset were described in detail in our previous publication (ref. 30), so we did not repeat this information in the current manuscript. For all additional markers used by Seurat for clustering, we have now provided the relevant information in the new Supplementary Table 1. Notably, Gata6 is one of the markers for MG duct and ductular cells.
Although the scRNA-seq was already published there are not yet available text tables with markers of each cluster/cell types. In this manuscript the authors should provide a Table with differentially expressed gene per cluster/celltypes in the supplemetary data. This would represent a valuable resource for the scientist interested in Meibomian gland and palpebral conjunctiva.
Response: We have provided the relevant information in Supplementary File 1 as requested.
While le manuscript is rich of interesting histological observations there is a general lack of quantification.
Response: We understand the reviewer’s concern. However, our major conclusions are based on clear yes-or-no determinations, and the differences in signal are readily distinguishable. Therefore, we believe that the current data are sufficient to support our conclusions.
Are goblet cells in the conjuntiva also progenies of Krt6a cells?
Response: This is an excellent point. We previously examined goblet cells but did not detect any Krt6a⁺ progeny within this population. However, our snRNA-seq dataset has revealed some interesting findings related to goblet cells. Because the differentiation and maintenance of goblet cells represent a distinct topic, we plan to publish these results in a separate study.
A schematic at the end of the manuscript, summarising the results could help the readers.
Response: We appreciate the reviewer’s valuable suggestion and have prepared a graphical abstract that summarizes the major conclusions of this study.
Reviewer 2 Report
Comments and Suggestions for Authors
The manuscript by Zhu et al. uses lineage tracing methods to determine the role of KRT6A and KRT17 as markers of distinct stem cell populations in Meibomian gland and palpebral conjunctiva. The manuscript is well-written with detailed methods. A few comments are listed below:
- Please provide total number of mice or human tissue used in the methods.
- Please provide age of human tissue used, since age is factor in MG dysfunction and disease.
- In Figure 2, it seems like the MG cells are also BrdU+. Please revisit the images.
- The authors should clearly state throughout all figures, whether each of the images are human or mouse tissue; and mouse or human gene names should follow correct nomenclature.
- Figure 3D is confusing because Krt6a-GFP+ cells are stained red, and KRT14 is stained green. At day 2, it appears to be expressed in superficial and suprabasal, but not basal cells. Please revisit the image.
- In Figure 5, it will be beneficial to also show the expression of KRT6A and KRT17 in human PC.
Author Response
Please provide total number of mice or human tissue used in the methods.
Because the total number of mice used for each assay varies, we stated in the Methods section that “At least three mice from each genotype and experimental condition were included in the assays.” For consistency, the number of human samples analyzed is also clarified in the corresponding figure legends. We have further emphasized this in the Methods section with the following statements: “The exact number of mice analyzed for each assay is provided in the corresponding figure legend.” and “The number of human eyelid samples used for histological analysis is specified in the figure legend.”
Please provide age of human tissue used, since age is factor in MG dysfunction and disease.
Response: We appreciate the reviewer’s valuable advice. The samples we analyzed were de-identified and discarded clinical specimens, and relevant demographic or clinical information was not available. Moreover, the limited number of samples makes it difficult to assess the potential influence of age on KRT6A and KRT17 expression in these tissues. We have addressed this limitation in the revised Discussion section as follows: “Second, it remains unclear whether clinical variables, such as age, sex, or cancer stage, affect KRT6A and KRT17 expression in human MG and MGC, which should be evaluated in a larger cohort.”
In Figure 2, it seems like the MG cells are also BrdU+. Please revisit the images.
Response: We agree with the reviewer that BrdU+ MG cells are present in the images. This is expected, as BrdU was administered via intraperitoneal injection, which labels all cells undergoing DNA synthesis, including proliferating MG cells. Because the turnover rate of murine MG cells has already been characterized in a previous study (PMID: 11340409 DOI: 10.1159/000055665), we did not specifically examine MG turnover in our work and instead focused on assessing the turnover rate of PC cells.
The authors should clearly state throughout all figures, whether each of the images are human or mouse tissue; and mouse or human gene names should follow correct nomenclature.
Response: We apologize for any confusion. All images presenting data from mouse experiments are accompanied by a schematic illustrating the experimental setup. For the only data involving human tissues (Figure 5H-K), the panels are clearly labeled as “Normal human MG” and “Human MGC” directly above the corresponding images. Additional clarification is also provided in the figure legend. We believe the current labeling is sufficiently clear. Regarding gene nomenclature, we have followed the standard guidelines throughout the manuscript and in all figures (https://www.jci.org/kiosk/publish/genestyle). However, if we have overlooked any inconsistencies, we would be happy to make the appropriate corrections.
Figure 3D is confusing because Krt6a-GFP+ cells are stained red, and KRT14 is stained green. At day 2, it appears to be expressed in superficial and suprabasal, but not basal cells. Please revisit the image.
Response: We understand the reviewer’s concern regarding the appearance of the GFP signal. Because the green channel generally exhibits higher background levels in IF assays than the red channel, we used an Alexa Fluor 555 secondary antibody to improve signal specificity. We believe the clear labeling in each panel is sufficient to indicate which signal corresponds to which marker. Regarding the GFP+ cells observed on Day 2, the reviewer is correct that most Krt6a-GFP+ cells are located in the superficial and suprabasal layers, which is consistent with endogenous Krt6a expression shown in Figure 1. However, we also observed sporadic Krt6a-GFP+ basal cells (pink arrow in Figure 3D), corresponding to the limited number of KRT6A-expressing basal cells (yellow arrow in Figure 1B). Therefore, the short-term lineage-tracing results recapitulated the endogenous Krt6a expression pattern, ensuring that the long-term tracing accurately reflects the true fate of these cells.
In Figure 5, it will be beneficial to also show the expression of KRT6A and KRT17 in human PC.
Response: We fully agree with the reviewer’s suggestion. We attempted to identify normal PC tissue in the available human eyelid samples; however, the PC tissue was either absent or obscured by tumor tissue, making it difficult to evaluate its expression. We are actively seeking additional human eyelid samples, but have not yet succeeded due to the limited availability of such specimens. We have addressed this limitation in the revised Discussion section as follows: “First, we did not examine KRT6A and KRT17 expression in human PC tissues because normal PC tissue was not available in the eyelid samples we analyzed. Assessing their expression in human specimens would help determine whether their expression patterns are conserved across species.”
Reviewer 3 Report
Comments and Suggestions for Authors
The manuscript by Zhu et al. is well written and presents the findings clearly. The topic is interesting and contributes valuable insights to the field. However, a few areas, as mentioned below, should be addressed to further enhance the clarity and overall impact of the manuscript.
Comments to the authors:
Major comments:
-In abstract, the conclusion is appropriate but would benefit from adding one sentence regarding the biological or clinical significance of identifying distinct stem cell populations in these tissues.
-The Introduction is generally well structured and provides a clear overview of tear film biology, Meibomian gland function, and conjunctival anatomy. However, this section is quite long and could be streamlined .
-Several paragraphs contain excessive background detail (e.g., functions of the tear film, meibocyte differentiation steps, mucin biochemistry). While informative, some of this information may not be essential for framing the current study and could be condensed.
-The authors clearly describe gaps in knowledge regarding Meibomian gland and conjunctival stem cells. It would be helpful to strengthen the link between these gaps and the rationale for investigating KRT6A and KRT17 specifically.
-The Introduction references many studies (e.g., HDAC3, KLF4, Hedgehog signaling) that regulate KRT6A and KRT17 in other tissues. While useful, some mechanistic details may be unnecessary at this stage and could be reserved for the Discussion.
-In Figures 3 and 4, it would strengthen the conclusions if the authors quantified the size or number of Krt6aGFP+ and Krt17GFP+ clones .
-For Figures 5–7, are the presented IF images representative of the entire tissue?
-Including quantitative data on the contribution of KRT6A+ basal cells to suprabasal layers in PC could better support the claim of local epithelial replenishment.
-The discussion is well-written, thorough, and effectively highlights the distinct roles of KRT6A and KRT17 in maintaining adult PC and MG homeostasis. The authors clearly integrate their findings with previous studies and provide mechanistic insights into regulation by Hh signaling, HDAC3, and KLF4.The tissue-specific effects of KLF4 deletion and the selective induction of KRT17 in palpebral conjunctiva (PC) versus Meibomian gland (MG) could be clarified further. Direct testing of promoter sensitivity for GLI2ΔN and the role of HDAC3 in repressing Hedgehog targets would strengthen mechanistic understanding.
-Discussing the functional impact of dysregulated KRT17 or KRT6A overexpression in ocular pathologies could enhance translational relevance.
-The expression of KRT6A and KRT17 in human PC remains unexplored. Addressing this limitation would strengthen the translational relevance of the findings.
Minor comments
-Authors are kindly requested to revise the references to conform to the journal’s formatting and citation style requirements.
-Authors are kindly requested to include an Abbreviations section in the manuscript.
- Provide the full term for MUC5AC in line 65 at its first appearance in the manuscript, followed by the abbreviation in parentheses.
-Similarly, please provide the full term for KLF4 in line 87 and check for others in the manuscript.
-Please include the IACUC number in the manuscript.
-Please clarify whether informed consent was obtained from donors or if consent was waived for the use of de-identified surgical waste tissue.
-Please provide a basic description of the BrdU pulse-chase assay.
- Please include the microtomy used to make the sections (Device name and company).
-Authors are kindly requested to consolidate all antibody information within the Immunofluorescence section to improve clarity and flow of the methods.
- Authors are kindly requested to include the number of mice used in the experiment and to clearly describe the method of sacrifice in the Materials and Methods section.
Thank you!
Author Response
-In abstract, the conclusion is appropriate but would benefit from adding one sentence regarding the biological or clinical significance of identifying distinct stem cell populations in these tissues.
Response: We appreciate the reviewer’s valuable advice and have added the following sentence at the end of the revised Abstract: “These findings provide important biological insight into tissue-specific maintenance mechanisms and may inform future therapeutic strategies for regenerating MG and PC tissues affected by SC exhaustion or dysregulation.”
-The Introduction is generally well structured and provides a clear overview of tear film biology, Meibomian gland function, and conjunctival anatomy. However, this section is quite long and could be streamlined.
Response: We agree with the reviewer that our Introduction is somewhat long. In previous submissions of our manuscripts and presentations of our data at various meetings, reviewers and audience members often advised us to provide more background to engage a broader readership, as this is a relatively specialized area. While senior experts, such as the reviewer, may feel that our background is lengthy, we have observed that many investigators new to this field are sometimes confused about topics such as tear film layers and dry eye disease classifications. A detailed Introduction, therefore, can help these readers better understand our study and related work. Given that there is no strict word limit, we believe providing this background enhances clarity and accessibility. We hope the reviewer can appreciate our rationale.
-Several paragraphs contain excessive background detail (e.g., functions of the tear film, meibocyte differentiation steps, mucin biochemistry). While informative, some of this information may not be essential for framing the current study and could be condensed.
Response: Please refer to the above response.
-The authors clearly describe gaps in knowledge regarding Meibomian gland and conjunctival stem cells. It would be helpful to strengthen the link between these gaps and the rationale for investigating KRT6A and KRT17 specifically.
Response: The rationale for investigating KRT6A and KRT17 in MG and PC stem cells is that KRT6A-expressing cells have been proposed as stem cells for the MG in a previous study, and KRT17 is expressed in the ductule, a MG compartment that houses stem cells, although their roles in PC stem cells remain unexplored. We have clarified this rationale in the Introduction as follows: “In the MG, our recent single-nucleus transcriptomic analysis revealed that Krt17 is expressed in the ductule, a region housing SCs that replenish MG ducts and acini. Furthermore, KRT6A localizes to label-retaining cells within ductules, suggesting that it may serve as a marker for self-renewing SCs. However, lineage tracing studies that track the fates of cells expressing KRT17 or KRT6A to determine whether they contribute to adult MG homeostasis have not been performed to date, and their functions in the PC also remain unexplored.” Please correct us if we have misunderstood the reviewer’s opinion, and we would be happy to modify the text accordingly.
-The Introduction references many studies (e.g., HDAC3, KLF4, Hedgehog signaling) that regulate KRT6A and KRT17 in other tissues. While useful, some mechanistic details may be unnecessary at this stage and could be reserved for the Discussion.
Response: We appreciate the reviewer’s advice and have condensed the relevant text as follows: “Expression of Krt6a and Krt17 is regulated by a variety of molecular mechanisms. These include Hh signaling, histone deacetylase 3 (HDAC3), and the transcription factor Krüppel-like factor 4 (KLF4) in various contexts.”
-In Figures 3 and 4, it would strengthen the conclusions if the authors quantified the size or number of Krt6aGFP+ and Krt17GFP+ clones.
Response: We appreciate the reviewer’s advice. However, our conclusion is a clear yes-or-no determination. Quantifying these parameters would not alter the core conclusion of our study.
-For Figures 5–7, are the presented IF images representative of the entire tissue?
Response: All the IF images in Figure 5-7 show representative results in the corresponding tissues.
-Including quantitative data on the contribution of KRT6A+ basal cells to suprabasal layers in PC could better support the claim of local epithelial replenishment.
Response: Performing this quantification is technically challenging. However, Figure 2E clearly shows the overall size and boundaries of Krt6a-GFP+ clones. Given the limited number and size of clones formed by Krt6a-GFP+ PC cells, we believe our data are sufficient to support their role in local tissue replenishment. Additionally, we have discussed this point in the manuscript as follows: “Notably, we did not observe stripes of Krt6a-GFP+ cells emanating from the corneal limbus, as would be expected if the limbus were the source of KRT6A+ stem cells; instead, KRT6A+ stem cells in the basal PC appear to locally replenish adjacent PC cells.”
-The discussion is well-written, thorough, and effectively highlights the distinct roles of KRT6A and KRT17 in maintaining adult PC and MG homeostasis. The authors clearly integrate their findings with previous studies and provide mechanistic insights into regulation by Hh signaling, HDAC3, and KLF4. The tissue-specific effects of KLF4 deletion and the selective induction of KRT17 in palpebral conjunctiva (PC) versus Meibomian gland (MG) could be clarified further. Direct testing of promoter sensitivity for GLI2ΔN and the role of HDAC3 in repressing Hedgehog targets would strengthen mechanistic understanding.
Response: We appreciate the reviewer’s valuable advice. GLI2ΔN (lacking the N-terminal) acts as an activator of Krt6a and Krt17 expression, and the promoter sensitivity of these genes to GLI2ΔN should be similar to that of full-length GLI2 (the N-terminal functions as a repressor while the C-terminal serves as an activator) as described in the cited reference. Regarding the mechanism by which HDAC3 represses Hh targets, we addressed this in the manuscript as follows: “HDAC3 deletion led to increased Hh pathway activity in the PC and MG, potentially contributing to the ectopic expression of KRT17 in Hdac3-deficient PC and MG acini. However, the precise mechanism by which HDAC3 suppresses Hh pathway activity in these tissues remains unclear.” We did not expand further on the regulation of the Hh pathway by HDAC3 because the primary focus of this study is the regulation of Krt6a and Krt17 expression.
-Discussing the functional impact of dysregulated KRT17 or KRT6A overexpression in ocular pathologies could enhance translational relevance.
Response: We fully agree with the reviewer’s opinion and have discussed this in the revised manuscript as follows: “Third, we did not investigate the functional roles of these stress keratins in PC and MG homeostasis or disease. Future studies using genetically modified mouse models to manipulate their expression could provide important mechanistic insights.”
-The expression of KRT6A and KRT17 in human PC remains unexplored. Addressing this limitation would strengthen the translational relevance of the findings.
Response: We appreciate the reviewer’s valuable suggestion and have included the following discussion in the revised manuscript: “First, we did not examine KRT6A and KRT17 expression in human PC tissues because normal PC was not available in the eyelid samples we analyzed. Assessing their expression in human specimens would help determine whether their expression patterns are conserved across species.”
Minor comments
-Authors are kindly requested to revise the references to conform to the journal’s formatting and citation style requirements.
Response: We have adjusted the reference format according to the journal’s guide.
-Authors are kindly requested to include an Abbreviations section in the manuscript.
Response: We have included an Abbreviation section as requested.
- Provide the full term for MUC5AC in line 65 at its first appearance in the manuscript, followed by the abbreviation in parentheses.
Response: We have modified the text accordingly.
-Similarly, please provide the full term for KLF4 in line 87 and check for others in the manuscript.
Response: We have modified the text accordingly. Additionally, the full term of all genes has been provided in the Abbreviation section.
-Please include the IACUC number in the manuscript.
Response: We have provided the IACUC number in the Method section.
-Please clarify whether informed consent was obtained from donors or if consent was waived for the use of de-identified surgical waste tissue.
Response: We have clarified this as follows in the revised manuscript: “The samples were de-identified specimens discarded from unrelated surgical procedures, and informed consent from patients was waived.”
-Please provide a basic description of the BrdU pulse-chase assay.
Response: We had described this in the original manuscript as follows: “This assay labels proliferative cells as they incorporate BrdU during DNA synthesis. Over time, the BrdU label is gradually diluted and eventually becomes undetectable after multiple rounds of cell division.”
- Please include the microtomy used to make the sections (Device name and company).
Response: We have specified this information in the Method section.
-Authors are kindly requested to consolidate all antibody information within the Immunofluorescence section to improve clarity and flow of the methods.
Response: We have made the corresponding modification in the revised manuscript.
- Authors are kindly requested to include the number of mice used in the experiment and to clearly describe the method of sacrifice in the Materials and Methods section.
Response: The number of mice used for each experiment is specified in the corresponding figure legend. The method of mouse euthanasia has been included in the revised Methods section.
Round 2
Reviewer 1 Report
Comments and Suggestions for Authors
Even if the snRNA-seq have been already published, I think that the umap representation on Fig.1 should be improved. For example I would show several umap representations of one or two genes that mark the relevant cell clusters. In addition the cluster numbering, till 33, on suppl. table 1 is not explained in the manuscript.
Author Response
Even if the snRNA-seq have been already published, I think that the umap representation on Fig.1 should be improved. For example I would show several umap representations of one or two genes that mark the relevant cell clusters. In addition the cluster numbering, till 33, on suppl. table 1 is not explained in the manuscript.
Response: We appreciate the reviewer’s suggestion. However, we would like to clarify that Fig. 1G presents feature plots displaying the expression patterns of Krt6a and Krt17 across all cell populations. Because more than 20 cell clusters are derived from the Meibomian gland and palpebral conjunctiva, including even a single signature gene for each cluster would result in an excessively large and overcrowded figure, ultimately reducing clarity.
All relevant information regarding cluster-specific marker genes has already been published in reference 30, and the complete list of signature genes is provided in Supplementary Table 1. We therefore believe that the current presentation is appropriate, while readers can readily locate the detailed gene information as needed.
Regarding the explanation for Supplementary Table 1, we have clarified the meaning of each column header in the revised manuscript as follows: “p_val indicates the p-value; avg_log2FC represents the average log₂ fold change of each gene; pct.1 denotes the proportion of cells within the cluster in which the gene is detected, whereas pct.2 reflects the proportion of cells across all other clusters in which the gene is detected. p_val_adj refers to the adjusted p-value; cluster indicates the numeric cluster identity; and gene lists the signature genes defining each cell cluster. ”
Reviewer 3 Report
Comments and Suggestions for Authors
The authors have satisfactorily addressed all my comments. I have no further comments and recommend that the manuscript be accepted for publication. Thank you.
Author Response
The authors have satisfactorily addressed all my comments. I have no further comments and recommend that the manuscript be accepted for publication. Thank you.
Response: We sincerely appreciate the reviewer’s recognition of our work and all the valuable suggestions to further improve our manuscript.